# An AGCRN Algorithm for Pressure Prediction in an Ultra-Long Mining Face in a Medium–Thick Coal Seam in the Northern Shaanxi Area, China

Xicai Gao [1,2,3,*] , Yan Hu [1,2,3,*], Shuai Liu [1,2,3], Jianhui Yin [4], Kai Fan [5] and Leilei Yi [5]

1   State Key Laboratory of Coal Resources in Western China, Xi'an University of Science and Technology, Xi'an 710054, China
2   Key Laboratory of Western Mine Exploitation and Hazard Prevention, Ministry of Education, Xi'an University of Science and Technology, Xi'an 710054, China
3   School of Energy, Xi'an University of Science and Technology, Xi'an 710054, China
4   Shaanxi Coal and Chemical Technology Institute Co., Ltd., Xi'an 710065, China
5   Sichuan Coal Industry Group Huarong Co., Ltd., Panzhihua 617000, China
*   Correspondence: gxcai07@163.com (X.G.); huu_yan@163.com (Y.H.)

**Abstract:** Due to the increase in the length of the mining face, the pressure characteristics and spatial distribution in fully-mechanized mining faces are different from those in typical mining faces, which leads to great challenges in roof management and the intelligent control of ultra-long mining faces. Taking the ultra-long mining face of a medium–thick coal seam in the northern Shaanxi mining area as an example and using field monitoring data for the working resistance of the hydraulic supports, a non-linear prediction method was used to extract the features of the dynamic data sequence of the working resistance of the hydraulic supports, and a deep learning method was used to establish a pressure prediction model for ultra-long mining faces based on the adaptive graph convolutional recurrent network (AGCRN) algorithm. In the proposed model, the supports in the fully mechanized mining face were regarded as the logic nodes of a topological structure, while the time-series resistance data for the supports were regarded as data nodes on a graph. The AGCRN model was used to determine the spatiotemporal relationship between the working resistance data of adjacent hydraulic supports, thereby improving the accuracy of the proposed model. The MAE and MAPE were employed as performance evaluation indices. When the node-embedding dimension was set to 10 and the time window was set to 16, the corresponding MAE and MAPE values of the prediction model were the minimum values. Compared with the reference models (i.e., the BP, GRU, and DCRNN models), the MAE and MAPE of the AGCRN model were 38.75% and 23.49% lower, respectively, indicating that the AGCRN model effectively demonstrates high accuracy in predicting the working resistance of supports. The AGCRN model was applied in the prediction of the working resistance of the supports of the ultra-long fully mechanized mining face. The results revealed that the working resistance of the supports in the lower and upper areas was relatively small along the strike, whereas the working resistance of the supports in the middle area was large, exhibiting a zoning pattern of "low-high-low" in terms of the average working resistance. In conclusion, the proposed model provides data references for the state of the hydraulic supports, pressure identification, and intelligent control of the ultra-long mining faces of the medium–thick coal seams in northern Shaanxi.

**Keywords:** ultra-long mining face; support resistance; deep learning; pressure prediction

## 1. Introduction

There are a large number of medium–thick coal seams in more than half of the main minable coal seam in the Yushen mining area in northern Shaanxi, and the geological conditions are relatively simple, with great excavation conditions. One of the most effective ways to reduce the drivage per ten thousand tons of coal, improve the productivity and

efficiency of the mining face, and reduce the loss of coal is to use ultra-long mining faces [1]. However, due to the increase in the length of the mining face, the behavior of mine pressure and its spatial distribution on a fully-mechanized mining face are different from those of typical mining faces. This results in great difficulties in roof management and the intelligent control of ultra-long mining faces [2–4]. Mine pressure behavior monitoring, pressure prediction and warning, and intelligent control are the keys to realizing the normal operation of the intelligent control system in ultra-long mining faces. By analyzing the working resistance of the supports of the ultra-long mining face and the pressure characteristics, deep learning algorithms have been used to achieve an accurate prediction of the pressure on the mining face, which is of great significance to realizing the safe, intelligent, and efficient mining of medium–thick coal seams.

In recent years, intelligent mining in China's coal mines has developed rapidly, and a number of intelligent mining faces (e.g., Huangling No. 2, Hongliulin, and Balasu Coal Mine) have been built. This has led to the intelligent sensing of strata information and the automatic control of key production systems and equipment in the mining face. Hydraulic supports are one of the main instruments used in the mining face. The posture, bearing capacity, and load distribution of supports represent the interaction between the support and the surrounding rock of the working face. State monitoring based on massive monitoring data for the support resistance of the mining face, including state sensing, analysis-discrimination, and autonomous adjustment, is the key to the intelligent control of the production system of the mining face [5].

Based on field monitoring data for support resistance in mining faces, a number of studies have investigated the pressure prediction in fully mechanized mining faces using methods such as the expert system, neural network, and big data analysis methods. For instance, back-propagation (BP) neural network models were constructed in several studies [6–8], and the performances of these models were verified using field monitoring data. At the same time, deep learning algorithms based on big data analysis have also been widely used [9–11], and great progress has been made in the establishment of a pressure-prediction model of the mining faces. Cheng et al. [12] extracted the characteristic parameters during the working cycle of the support and constructed an intelligent sensing system for the supports and roof of a fully- mechanized mining face, achieving a quality evaluation of the support and an intelligent prediction of roof weighting. Zhao et al. [13] used a relational database to store working resistance data and extracted the time-series characteristics of the hydraulic supports during the advancement of the mining face. Then, they employed a long-short-term memory network (LSTM) to establish a pressure-prediction model. With the help of transfer learning, they proved that the LSTM model had a good generalization ability in roof-weighting prediction. Based on the time-series distribution of the working resistance of a support and the characteristics of complex working conditions, Pang et al. [14] developed a classification modeling method for support loading based on a clustering algorithm. Additionally, Zeng et al. [15] established a Prophet + LSTM model for pressure prediction on a mining face by integrating the working resistance data for adjacent supports using additional regression variables models. Compared to previous individual models, the combined Prophet + LSTM model better captured the composite features of the time-series data and achieved roof pressure prediction during the advancement of the mining face.

Working resistance monitoring data for the supports of a fully mechanized mining face provide a data basis for realizing the intelligent sensing of the performance of the supports and the stability of the roof [16]. The working resistance of the supports is dynamic and is affected by many factors, which leads to great challenges in pressure prediction for a mining face. A novel idea is to improve the pressure-prediction model accuracy by mining the characteristics of the massive monitoring data themselves for a mining face and applying non-linear prediction methods. The graph neural network has great advantages in extracting and analyzing the associations between nodes and in reflecting the topological relationships between objects [17–19]. The supports in the fully mechanized

mining face were regarded as the logic nodes on a topological structure, and the time-series resistance data for the supports were regarded as data nodes on a graph. The adaptive graph convolutional recurrent network is utilized to extract the spatiotemporal correlation of the resistance data for adjacent supports, which effectively improves the accuracies of pressure-prediction models.

Therefore, in this study, considering the influence of the working resistance of adjacent hydraulic supports in the ultra-long mining face of a medium–thick coal seam, an adaptive graph convolutional recurrent network (AGCRN) was constructed to determine the spatiotemporal relationship between the working resistance of adjacent hydraulic supports in the mining face, thereby establishing an ultra-long mining face pressure prediction model. On this basis, a roof-weighting early warning model was constructed for pressure analysis and early warning. Engineering verification revealed that the early warning system has good performance. The results of this study provide a data reference for the status monitoring, pressure monitoring, and intelligent control of the hydraulic supports in the ultra-long mining faces of the medium–thick coal seams in northern Shaanxi.

## 2. Working Resistance Prediction Model for Supports in Ultra-Long Mining Faces

The working resistance of supports during the advancement of fully mechanized mining faces is expressed as dynamic time-series data, and the data for different supports are correlated in the time and space domains. A support is constantly affected by its neighboring supports and other supports while it is reaching equilibrium. In this study, the graph theory was employed to reflect the topological structure between the hydraulic supports; the node-adaptive parameter learning (NAPL) module and the data-adaptive graph generation (DAGG) module were used to construct the graph; and the graph neural network was applied to extract the spatiotemporal correlation features of the working resistance data for the supports. Ultimately, an adaptive graph convolutional recurrent network (AGCRN) was developed for pressure prediction.

### 2.1. Adaptive Graph Convolutional Recurrent Network

The AGCRN [20,21] was composed of two modules: the NAPL and DAGG. The architecture of the model is illustrated in Figure 1.

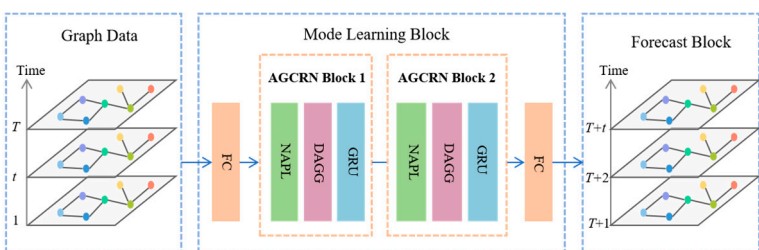

**Figure 1.** Flow diagram of the AGCRN model.

The node-embedding matrix and the weight pooling matrix were introduced in the NAPL module to change the parameter sharing mode of the original graph convolutional neural network, and the DAGG module was utilized to capture the hidden relationship between different sequences and to obtain the adjacency information.

### 2.1.1. NAPL Module

The graph convolution module of traditional graph convolutional networks (GCNs) usually uses first-order Chebyshev Polynomials:

$$Z = \left( I_N + D^{-1/2} A D^{-1/2} \right) X \Theta + b \tag{1}$$

$$A \in R^{N \times N}, X \in R^{N \times C}, \Theta \in R^{C \times F}, b \in R^F \tag{2}$$

where $Z$ is the graph representation learned by the GCN; $A$ and $D$ are the adjacency matrix and degree matrix of a graph containing $N$ nodes, respectively; $X$ and $\Theta$ are the input feature matrix with dimension $C$ and the output feature matrix with dimension $F$, respectively; $\Theta$ and $b$ are the parameters to be learned by the GCN from the data.

The nodes in the graph are different from each other, and the $N$ nodes have $N$ learnable parameter spaces. In this case, $\Theta \in R^{N \times C \times F}, b \in R^{N \times F}$. When the number of nodes is large, overfitting easily occurs. By introducing node embedding and matrix decomposition, the NAPL module is decomposed $\Theta$ into a node-embedding matrix $E_G \in R^{N \times d}$ and a weight pooling matrix $W_G \in R^{d \times C \times F}$. Moreover, the parameter $b$ is decomposed into a node-embedding matrix $E_G \in R^{N \times d}$ and a weight-pooling matrix $b \in R^{d \times F}$. Thus, the graph convolutional neural network becomes:

$$Z = \left( I_N + D^{-\frac{1}{2}} A D^{-\frac{1}{2}} \right) X E_G W_G + E_G b_G \tag{3}$$

### 2.1.2. DAGG Module

In traditional GCN-based prediction models, a predefined adjacency matrix $A$ is usually required for convolutional operations. However, the predefined matrix is subjective and cannot reflect the complete spatial information of the nodes in the graph. The DAGG module regards the normalized adjacency matrix as a learnable parameter, and the equation is as follows:

$$D^{-\frac{1}{2}} A D^{-\frac{1}{2}} = softmax \left( ReLU \left( E_A \cdot E_A^T \right) \right) \tag{4}$$

where $E_A \in R^{N \times d_e}$ is a randomly initialized node-embedding matrix; $d_e$ is the dimension of node embedding; and $E_A E_A^T$ is the similarity between nodes.

$E_A$ is the inner product of the embedding vectors of node $i$ and node $j$ and it can learn the hidden dependencies between different sequences autonomously during the training process. The GCN improved by the DAGG module is as follows:

$$Z = \left( I_N + softmax \left( ReLU \left( E_A \cdot E_A^T \right) \right) \right) X \Theta \tag{5}$$

The gated recurrent unit (GRU) is used in the AGCRN to capture the time dependence, and the MLP layer of the GRU is replaced with the graph convolutional neural network learned by the NAPL module. The formula is as follows:

$$\widetilde{A} = softmax \left( ReLU \left( EE^T \right) \right) \tag{6}$$

The update gate $z_t$ in the GRU model determines the extent to which the state information at time $t$-1 is passed to the current state at time $t$; and the reset gate $r_t$ is employed to control the extent to which the state at time $t$ is passed to the current hidden state $\widetilde{h}_t$. The GRU update formula optimized by the AGCRN is as follows:

$$z_t = \sigma \left( \widetilde{A} [X_{:,t}, h_{t-1}] E W_Z + E b_Z \right) \tag{7}$$

$$r_t = \sigma \left( \widetilde{A} [X_{:,t}, h_{t-1}] E W_r + E b_r \right) \tag{8}$$

$$\hat{h}_t = tanh \left( \widetilde{A} [X_{:,t}, r \odot h_{t-1}] E W_{\hat{h}} + E b_{\hat{h}} \right) \tag{9}$$

$$h_t = z_t \odot h_{t-1} + (1 - z_t) \odot \hat{h}_t \tag{10}$$

where $\sigma$ is the sigmoid function; $X_{:,t}$ and $h_t$ are the input and output in time step $t$; $h_{t-1}$ is the output in time step $t-1$; is the hidden state; [.] is the connection operation; and $\Theta$ is the Hadamard product. $W_z$, $b_z$, $W_r$, $b_r$, $W_{\hat{h}}$, and $b_{\hat{h}}$ are the learnable parameters in the AGCRN. The structure of the AGCRN is shown in Figure 2.

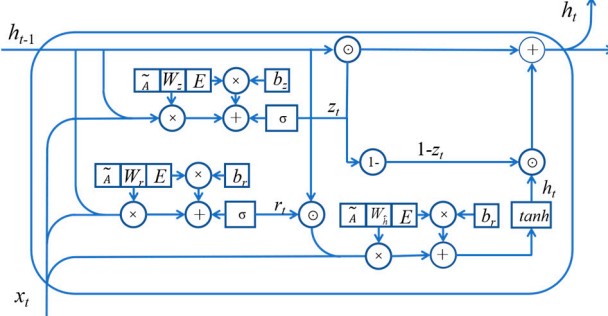

**Figure 2.** Structure of the AGCRN model.

*2.2. Support Pressure-Prediction Model*

2.2.1. Modeling Process

With the cyclic lowering and uplifting of supports during the advancement of a fully-mechanized mining face, the load of the overlying strata on the mining face will vary in different regions and between different supports along the dip direction. The resistance variation of the target support and adjacent supports is similar. Hence, to predict the support resistance of the mining face, the AGCRN model can be employed to obtain the dynamic correlation between the resistances of the supports based on the NAPL module. The overall flow of the model is presented in Figure 3 and described below.

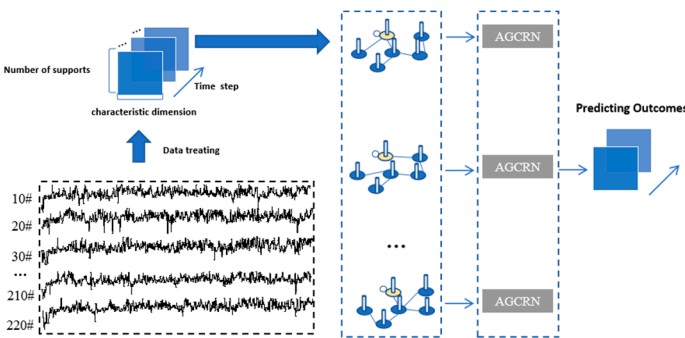

**Figure 3.** Flow chart of roof-weighting prediction.

(1) Dataset construction. The dataset utilized was the end-of-loop resistance data of the support during the mining of the #202 mining face of a coal mine in Shaanxi from July to December 2021. The mining height of the working face is 1.8–2.6 m. The data frequency was the mining cycle/time. The end-of-loop resistance of the support reflects the maximum load during each mining cycle and can characterize the bearing capacity of the support and the breakage of the roof.

(2) Data preprocessing. Abnormalities such as data loss, outliers, and noise are often encountered in the monitoring, analysis, and data transmission processes. To ensure data quality, it was necessary to preprocess the raw data. Outliers occurred during circumstances in which there were short or long intervals of data collection. In this study, the outliers of the support working resistance with a duration of less than 20 min or longer than 3 h were eliminated. The missing values of the support working resistance are generally due to data loss during transmission or sensor errors. For the missing values, the Lagrange interpolation method was applied to fill the gaps.

(3) Time dimension unification. The dataset contained multiple time series and was converted into a graph structure. The amount of original data was huge and heterogeneous, and the monitoring time points for the different supports were different. It is very difficult to analyze the spatial relationship without a unified time scale. To unify the time dimension of the data, the end-of-loop resistance data for the day were averaged and employed as

the data for the day in the extraction process. The extraction process of support resistance features is shown in Figure 4.

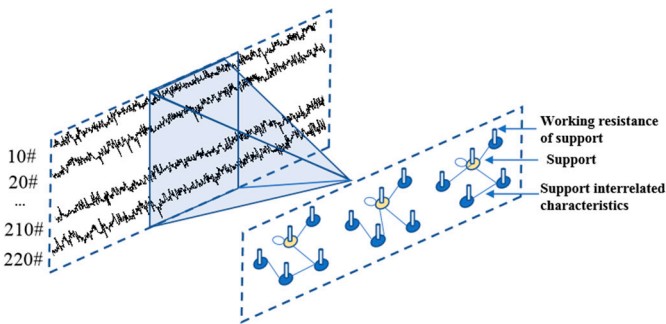

**Figure 4.** Feature extraction from the support working resistance data.

(4) Data normalization. Normalization is generally used to adjust the interval of the data proportionally, such that the data are converted into the same value range. Data normalization can effectively reduce the impact of excessive eigenvalues in the original data and can reduce training errors. In this study, the min-max method was applied to normalize the original data to the range of [0,1]. The formula is as follows:

$$\overline{x} = \frac{x - min(x)}{max(x) - min(x)} \tag{11}$$

where $x$ is the original resistance data; $\overline{x}$ is the normalized resistance data; and max and min are the maximum and minimum values in the original resistance data, respectively.

(5) Training set and testing set. It is necessary to divide the dataset into a training set and a testing set. During the construction of the model, it is also essential to check the degree of fitting and to select the optimal hyperparameters. Hence, the training data were further divided into a training set and a verification set. After training, it was necessary to objectively evaluate the generalization ability of the model, and the verification set was used as the testing set. The original resistance data were divided according to a ratio of 7:3.

(6) Training and verification. The training set was utilized to optimize the model and determine the best hyperparameters, and the testing set was employed to conduct the prediction. The mean absolute error (MAE) and mean absolute percentage error (MAPE) were used as the performance indices to evaluate the model's accuracy.

### 2.2.2. Parameter Setting

The algorithm in this study was implemented in the PyTorch framework. The basic environment parameters are listed in Table 1.

**Table 1.** Experimental parameters.

| Experimental Environment | Parameter |
| --- | --- |
| Operating system | Windows 10 |
| Development tool | PyCharm |
| CPU | i5-4210U |
| GPU | GeForce 950M |
| CUDA | 10.2 |
| Internal storage | 4 G |
| Python | 3.7 |

The performance evaluation indices are described below.

(1) MAE

The MAE represents the average value of the error between the predicted value and the measured value, which can avoid the problem of mutual cancellation of errors

and therefore, accurately reflect the actual error. The smaller the MAE, the better the performance of the model. The calculation formula is as follows:

$$\text{MAE} = \frac{1}{n}\sum_{i=1}^{n}|y_i - \hat{y}_i| \tag{12}$$

where $n$ is the sample size, $y_i$ is the monitoring data, $\hat{y}_i$ is the predicted data, and $\overline{y}$ is the mean value of the sample points.

(2) MAPE

The MAPE is the average percentage error between the predicted value and the measured value. It is sensitive to changes in the relative error and can accurately reflect the model error. The smaller the MAPE, the better the performance of the model. The calculation formula is as follows:

$$\text{MAPE} = \frac{100\%}{n}\sum_{i=1}^{n}\left|\frac{y_i - \hat{y}_i}{y_i}\right| \tag{13}$$

where $n$ is the sample size, $y_i$ is the monitoring data, and $\hat{y}_i$ is the predicted data.

The end-of-loop resistance data for the supports in the #202 mining face were used, and the data were divided into a training set and a testing set according to a ratio of 7:3. The Adam optimization algorithm was employed, with an initial learning rate of 0.001 and a convolution kernel size of 4. The number of model layers was set to 2, and the number of nodes in the hidden layer was set to 64. Since the training and prediction data were both end-of-loop resistance data for the supports, the input and output dimensions of the model were both 1, and the number of epochs was set to 200. The Early Stop mechanism was added to the model, which can effectively prevent the model's generalization performance from deteriorating by limiting the number of training iterations of the model in order to minimize the loss function.

### *2.3. Results and Analysis*

2.3.1. Hyperparameter Optimization

(1) Node-embedding dimension optimization

A key parameter in the AGCRN model is the node-embedding dimension. The optimal dimension for node embedding in the model is determined through hyperparameter optimization. The model performance under different node-embedding dimensions is shown in Figure 5.

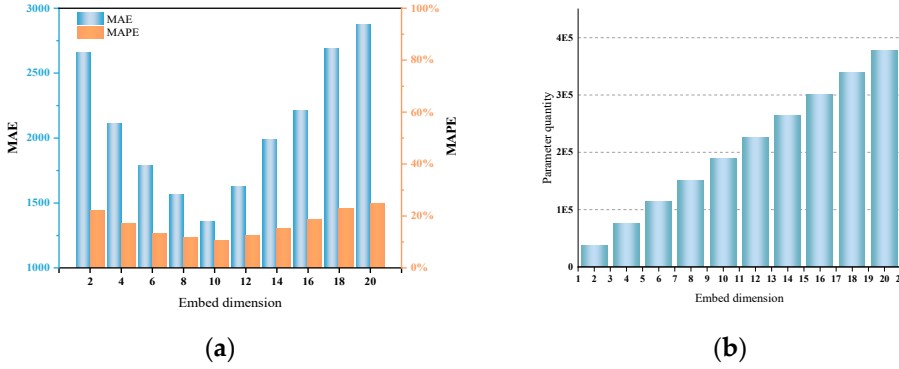

(**a**)                    (**b**)

**Figure 5.** Performance impact graph of embed dimension, they should be listed as: (**a**) Embedding dimension error effect; (**b**) Number of embedded dimension parameters.

As shown in Figure 5a, the MAE and MAPE values of the model initially decreased with increasing node-embedding dimension, and then gradually increased. When the embedding dimension was 10, the MAE was 1358.21, the MAPE was 10.45%, and the error was the minimum value. As can be seen from Figure 5b, the number of parameters in the

model exhibited a linear growth trend as the embedding dimension increased. The results indicate that a high-dimensional node-embedding matrix contained more information, which helped the DAGG module to infer more spatial correlation information. Moreover, a higher dimension significantly increased the number of parameters of the NAPL module, making it more difficult to optimize the model. Therefore, when the node-embedding dimension was 10, the error of the AGCRN was the minimum value, and the performance was optimal.

(2) Time window optimization

When predicting time series, historical data are usually used to predict the data at the current moment, and the size of the historical data is referred to as the time window. If the time window is too small, the historical data contain incomplete and insufficient information, which affects the prediction results. However, if the time window is too large, the model may not be able to accurately grasp the temporal relationship between the data. Therefore, it is essential to choose an optimal time window. The time window was set to 8, 16, 32, and 64 days, and the other parameters of the model were not changed. The MAE and MAPE were compared. The results are presented in Table 2.

**Table 2.** Model performance under varying time windows.

| Time Window (Days) | MAE | MAPE |
|:---:|:---:|:---:|
| 8 | 1497.31 | 13.05% |
| 16 | 1351.47 | 11.68% |
| 32 | 1948.21 | 15.86% |
| 64 | 2663.28 | 20.49% |

As the time window increased, the model error initially decreased and then increased. When the time window was 16, the MAE and MAPE values reached the minimum values. As the time window continued to increase, the model error increased further. Therefore, in this study, the time window was set to 16 days to predict the working resistance of the supports during the next 3 days.

### 2.3.2. Performance Comparison and Analysis

(1) Selection of supports for prediction

As shown in Figure 6, the supports in the fully mechanized mining face were divided into three groups, i.e., upper, middle, and lower measurement areas, along the strike. During the advancement of the mining face, the end-of-loop resistances of the adjacent supports exhibited similar characteristics, and the amplitude and range were highly coupled. Therefore, it was necessary to perform the prediction based on the working resistance of the highly correlated adjacent supports [22].

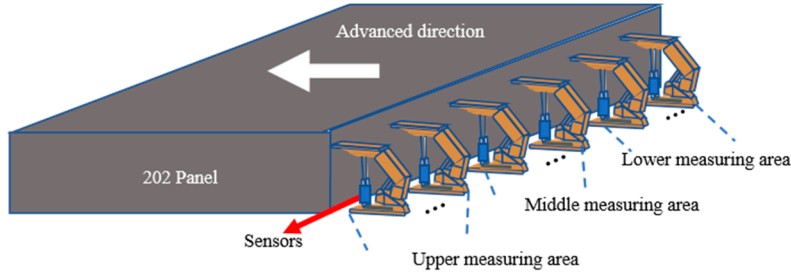

**Figure 6.** Classification of supports in the #202 fully- mechanized mining face.

Compared with the Pearson correlation coefficient, the maximum information coefficient (MIC) [23] can better characterize the degree of correlation between two sequences. Therefore, the MIC was utilized to measure the correlation between the working resistances

of the supports in the mining face, and the support with the greatest correlation with the remaining supports in each measurement area was used as the predictive support. The work resistance time series of two supports, $X_i$ and $X_j$, were divided into $x \times y$ segments.

$MIC(X_i; X_j)$ was calculated as follows:

$$mic\left(X_i; X_j\right) = \max_{x*y<B} \frac{\int P(x,y)log_2 \frac{P(x,y)}{P(x)P(y)}dxdy}{log_2 min(x,y)} \tag{14}$$

where $P(x,y)$ is the joint probability distribution of $x$ and $y$. $P(x)$ and $P(y)$ are the probability distributions of $x$ and $y$, respectively. $B$ is the maximum resolution.

The sum of the MICs of the working resistance of support $i$ and those of the other supports was calculated as follows:

$$mic^i(x;y) = \sum_{j=1}^{N} mic\left(X_i; X_j\right) \tag{15}$$

The support with the maximum MIC $mic^i_{max}(x;y)$ in each measurement area was selected as the predictive support. Due to space limitations, only the details for the support in the lower measurement area are presented (Table 3).

**Table 3.** MIC values of supports in the lower measurement area.

| Support No. | #10 | #20 | #30 | #40 | #50 | #60 | #70 |
|:---:|:---:|:---:|:---:|:---:|:---:|:---:|:---:|
| MIC | 5.5198 | 5.3124 | 6.4155 | 6.2083 | 5.8840 | 6.6877 | 6.4357 |

Among the supports in the lower measurement area, support #60 had the largest MIC value (6.6877), indicating that the correlation between support #60 and the other supports in the lower measurement area was the highest. Thus, support #60 was selected in the testing set as the representative support of the lower measurement area and was employed to compare the performance of the AGCRN model and the reference models, thereby verifying the performance of the AGCRN model.

(2) Performance comparison

Figure 7 and Table 4 present a comparison of the original value and the results for support #60 predicted using four different models. The $x$-axis is the samples in the testing set, and the $y$-axis is the time-weighted average end-of-loop resistance.

**Table 4.** Performances of the four reference models in pressure prediction of the #60 support in the lower measurement area.

| Models | MAE | MAPE |
|:---:|:---:|:---:|
| BP | 1322.78 | 10.69% |
| GRU | 1116.42 | 9.02% |
| DCRNN | 797.34 | 6.47% |
| AGCRN | 585.15 | 6.19% |

As shown in Table 4, the MAE values of the BP model, GRU model, and DCRNN model were 797.34–1322.78, and the MAPE values were 6.47–10.69%. In comparison, the MAE of the AGCRN model was 585.15, and the MAPE was 6.19%. The errors of the DCRNN and AGCRN models were significantly lower than those of the other single-sequence prediction models, and the AGCRN had the smallest prediction error, demonstrating the effectiveness of the proposed AGCRN model.

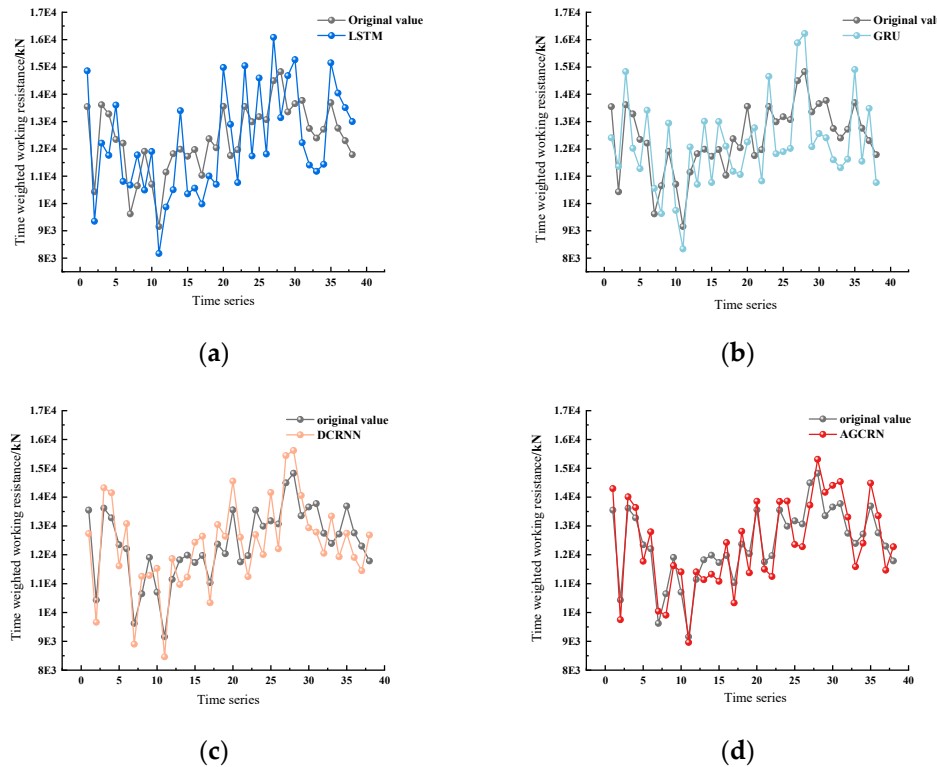

**Figure 7.** Comparison of the performances of the AGCRN model and four reference models, they should be listed as: (**a**) LSTM; (**b**) GRU; (**c**) DCRNN; (**d**) AGCRN.

## 3. Sub-Zoning Pressure Prediction Results for the Supports in the Ultra-Long Mining Face

### 3.1. Prediction of Support Resistance

Based on the resistance monitoring data for the supports in an ultra-long fully mechanized mining face in Shaanxi, a roof-weighting criterion value was generated according to the historical data collected during the advancement of the mining face. Then, the AGCRN model was applied to predict the changes in the working resistance of the supports in the #202 mining face. When the predicted resistance value of the support exceeded the threshold, the ultra-long mining face was considered to be under roof weighting. The prediction results serve as valuable guidance for whether production dispatchers should promptly execute an emergency response plan.

Due to space limitations, only support #60 in the lower measurement area is discussed to illustrate the results.

Figure 8 shows the predicted working resistance of support #60 in the lower measurement area of mining face #202. The green line represents the measured end-of-loop resistance of the support, the red line represents the predicted value of the end-of-loop resistance of the support based on the AGCRN model, and the yellow line represents the value of the roof-weighting criterion of the ultra-long fully mechanized mining face. The errors in the predicted working resistance and roof-weighting interval were both within 5%, suggesting that the AGCRN model can achieve the effective prediction of the resistance of the support and the roof weighting of the mining face.

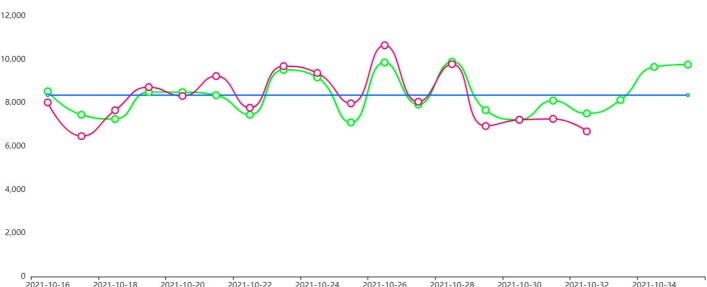

**Figure 8.** Comparison of the predicted and measured working resistance of the #60 support.

### 3.2. Analysis of Support Resistance in Different Areas of the Ultra-Long Mining Face

(1) Variation in support resistance with advancement distance in different areas.

The supports in the #202 ultra-long mining face were divided into three groups along the strike, i.e., lower (#10–#30), middle (#40–#180), and upper (#190–#220) groups. The prediction results revealed the zoning characteristics of the ultra-long fully mechanized mining face.

Figure 9 presents the working resistances of the representative supports. The average resistance of the lower supports (#10–#30) was 9723.17–10,141.65 kN, suggesting a low overall resistance. The average working resistance of the middle supports (#50–#140) was 11,423.47–13,016.48 kN, indicating a high resistance and large variations. The average working resistance of the upper supports (#200–#220) was 9821.96–11,151.89 kN, implying a low working resistance. In conclusion, during the advancement of the ultra-long fully mechanized mining face, the working resistance of the supports had obvious zoning characteristics along the strike and exhibited a low-high-low pattern.

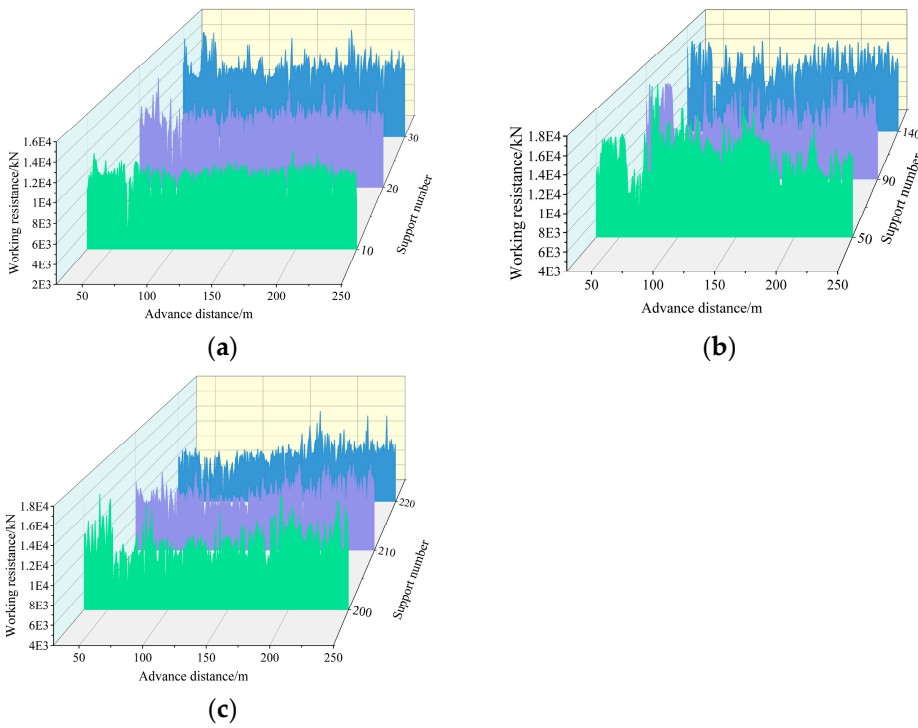

**Figure 9.** Working resistance distribution characteristics of different areas along the strike, they should be listed as: (**a**) lower measuring area (#10–#30); (**b**) middle measuring area (#40–#180); (**c**) upper measuring area(#190–#220).

(2) Zoning characteristics of the support resistance under the mining face under roof weighting.

As shown in Figure 10a, when the mining face was first subjected to roof weighting, the average working resistance of the middle supports (#40–#90) was 15,598.39 kN, and the average working resistance of supports #130–#180 was 15,402.35 kN. The resistance of the lower and upper supports was low; the average resistance of supports #10–#30 was 12,335.15 kN, and that of supports #190–#220 was 12,155.7 kN. As can be seen from Figure 10b, when the mining face was first subjected to periodic roof weighting, the average resistance of the middle supports (#30–#90) was 15,549.95 kN, and that of supports #14–#190 was 15,915.20 kN. The working resistance of the lower and upper supports was low. The average resistance of supports #10–#20 was 11,698.79 kN, and that of supports #200–#220 was 12,505.33 kN. In contrast to mining faces with a normal length, the working resistance of the supports along the ultra-long mining face was high in the lower and upper parts of the middle area (i.e., #40–#80 and #120–#180) during the periodic roof weighting, exhibiting a low-high-medium-high-low distribution pattern along the strike.

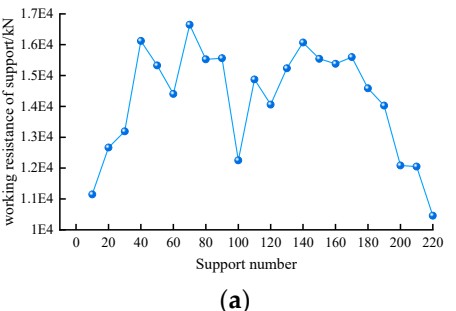
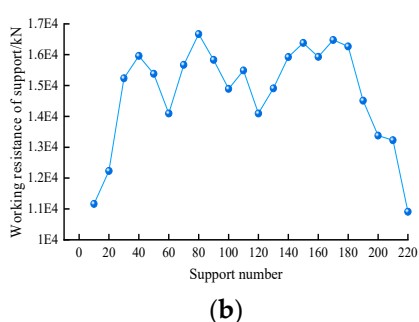

(**a**)  (**b**)

**Figure 10.** Working resistance distribution in different areas under the condition of roof weighting, they should be listed as: (**a**) Initial roof weighting; (**b**) First-cycle roof weighting.

## 4. Conclusions

(1) Based on working resistance data for supports recorded during the mining of a fully mechanized mining face in the Yushen mining area in northern Shaanxi, the NAPL module and DAGG module in the AGCRN model were used to dynamically extract the spatiotemporal correlation between the resistance data for adjacent supports in the mining face. An AGCRN prediction model was developed to predict the working resistance of the supports in fully mechanized mining faces.

(2) The MAE and MAPE were employed as performance evaluation indices. When the node-embedding dimension was set to 10 and the time window was set to 16, the corresponding MAE and MAPE values of the prediction model were the minimum values.

(3) Three reference models (i.e., the BP, GRU, and DCRNN models) were selected and compared with the proposed AGCRN model. The results reveal that compared with the reference models, the MAE and MAPE of the AGCRN model were 38.75% and 23.49% lower, respectively, indicating that the AGCRN model effectively captures spatiotemporal correlation information and demonstrates high accuracy in predicting the working resistance of supports.

(4) The AGCRN model was applied to the prediction of roof weighting in an ultra-long fully mechanized mining face in the Yushen mining area in northern Shaanxi. The results show that the working resistance of the supports in the lower and upper areas was relatively small along the strike, whereas the working resistance of the supports in the middle area was large, exhibiting a zoning pattern of "low-high-low" in terms of the average working resistance. During the periodic roof weighting, the working resistance curve of the supports in the middle area displayed a double-peak pattern. The overall working resistance of the fully mechanized mining face had a low-high-medium-high-low trend along the strike. We compared the predicted results with the field-measured data and related literature and found that the predicted model has good accuracy. In the next

step, we will combine the numerical analysis results and continuously optimize the model parameters to expand the application range and effect of the model.

**Author Contributions:** Conceptualization, X.G., Y.H. and J.Y.; methodology, Y.H. and K.F.; software, S.L. and L.Y.; validation, J.Y., X.G. and K.F.; formal analysis, X.G. and L.Y.; investigation, L.Y.; resources, K.F.; data curation, S.L., K.F. and L.Y.; writing—original draft preparation, X.G., Y.H. and J.Y.; Writing-review and editing, X.G. and K.F.; visualization, S.L.; supervision, K.F.; project administration, L.Y.; funding acquisition, K.F. and L.Y. All authors have read and agreed to the published version of the manuscript.

**Funding:** This research was funded by the Natural Science Basic Research Program of Shaanxi (Program No. 2021JLM-10), Shaanxi Provincial Outstanding Youth Science Foundation Project (2023-JC-JQ-42), The Youth Innovation Team of Shaanxi Universities. The APC was funded by the Natural Science Basic Research Program of Shaanxi (Program No. 2021JLM-10).

**Institutional Review Board Statement:** Not applicable.

**Informed Consent Statement:** Not applicable.

**Data Availability Statement:** The data used to support the findings of this study are included in the article.

**Conflicts of Interest:** The authors declare no conflict of interest.

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
