# Peer review of "An AGCRN Algorithm for Pressure Prediction in an Ultra-Long Mining Face in a Medium–Thick Coal Seam in the Northern Shaanxi Area, China"

_applsci, doi:10.3390/app132011369_

Round 1
Reviewer 1 Report
Comments and Suggestions for Authors
I have two suggestions to authors
- Please indicate the limitation of medium-thick coal seams
- Measured data should be given in text
Author Response
1.Please indicate the limitation of medium-thick coal seams
RESPONSES: Thanks for your reminder. According to the general classification method of coal seam thickness, less than 1.3m is a thin coal seam, and more than 3.5m is a thick coal seam. The thickness of the coal seam in this paper is 2m-3m, which is called a medium-thick coal seam. And The mining height of the #202 working face is 1.8-2.6m. The mining height was added in line 196-197.
2. Measured data should be given in text
RESPONSES: Thanks for your reminder. Due to the limitation of the length of the paper, the large amount and variety of measured data are not added in the paper, and we can provide it from the correspondence author on reasonable request.
Reviewer 2 Report
Comments and Suggestions for Authors
The authors proposed a pressure prediction model for ultra-long mining faces based on the adaptive graph convolutional recurrent network (AGCRN) algorithm of a medium-thick coal seam, which was trained on the ultra-long mining face of a medium-thick coal seam in the northern Shaanxi mining area as an example and using field monitoring data for the working resistance of the hydraulic supports. The topic is interesting, and the novelty of the work and contributions of the authors are substantial. The numerical examples support the methodology and illustrate the applicability of the model; however, the following points need to be implemented /clarified:
1. First, the introduction section could be improved to mention and refer to the recent contribution of deep learning in Engineering with Computers: https://www.sciencedirect.com/science/article/abs/pii/S0898122123002122
​https://link.springer.com/article/10.1007/s11440-021-01440-1 ​https://link.springer.com/article/10.1007/s00366-021-01586-2
2. The proposed graph convolutional recurrent network (AGCRN) algorithm were compared with LSTM, GRU, DCRNN and AGCRN. There are other classical and state-of-the-art deep learning model fitted to deal with time-series data such as bidirectional long short-term memory (BiLSTM), Bidirectional GRU and Time Series Transformer, please try to add more predictive models in comparison.
3. The readability and presentation of the study should be further improved. Please proofread the whole manuscript to improve the language and correct those typos and grammar errors. Please adjust the size of figures, tables, and reference to match with the width of the manuscript.
4. The hyperparameter settings are important for readers to reestablish the model, which should be introduced. Please show the hyperparameters in tables for adopted machine learning algorithms and those models in comparison. Also, please add an introduction to the hyperparameter tuning and selection procedure that ensures a fair evaluation and comparison.
5. This application is a pure data-driven model; however, the authors did not give too much introduction to data mining with the data. The authors should add an introduction to the data interpolation and feature engineering.
6. How do you ensure that you have ‘good’ data in your approach? Or more precisely: How do you ensure that no overfitting occurs? The results on both the training and validation dataset and learning curves are missing, which will help us evaluate if the data-driven model is overfitting. Please add in the revision.
7. The model evaluation is not clearly introduced. To prevent overfitting, a K-fold cross-validation technique should be adopted to help evaluate the model to avoid a high variance and bias in the results generated. Besides, one sampling comparison is not convincing enough, you need to show the average results of RMSE for a repeated sub-sampling process, that’s why repeated cross-validation must be added here. Please add in the revision.
Comments on the Quality of English LanguageThe readability and presentation of the study should be further improved. Please proofread the whole manuscript to improve the language and correct those typos and grammar errors. Please adjust the size of figures, tables, and reference to match with the width of the manuscript.
Author Response
1.First, the introduction section could be improved to mention and refer to the recent contribution of deep learning in Engineering with Computers: https://www.sciencedirect.com/science/article/abs/pii/S0898122123002122 https://doi.org/10.1007/s11440-021-01440-1 https://doi.org/10.1007/s00366-021-01586-2
RESPONSES: Thanks for your advice. The recent contribution of deep learning in Engineering with computers were added in the introduction section line77-79 and references 454-459.
2. The proposed graph convolutional recurrent network (AGCRN) algorithm were compared with LSTM, GRU, DCRNN and AGCRN. There are other classical and state-of-the-art deep learning model fitted to deal with time-series data such as bidirectional long short-term memory (BiLSTM), Bidirectional GRU and Time Series Transformer, please try to add more predictive models in comparison.
RESPONSES: Thank you, your suggestion is valid. For time series data, we refer to relevant literature and have conducted relevant comparisons on adaptability and accuracy when selecting deep learning models. However, due to space limitations, they were not reflected in the text.Meanwhile, we feel that the scope of work of the present paper can support its conclusions. Therefore, we will consider it in future papers in a follow-up paper.
3. The readability and presentation of the study should be further improved. Please proofread the whole manuscript to improve the language and correct those typos and grammar errors. Please adjust the size of figures, tables, and reference to match with the width of the manuscript.
RESPONSES: Thank you very much for finding this error. We are sorry for this grammar problem and have corrected it according to your suggestion. The sizes of figures, tables and reference to match are adjusted according to the width of the manuscript. In addition, we have asked a native English editor to revise the manuscript.
4. The hyperparameter settings are important for readers to reestablish the model, which should be introduced. Please show the hyperparameters in tables for adopted machine learning algorithms and those models in comparison. Also, please add an introduction to the hyperparameter tuning and selection procedure that ensures a fair evaluation and comparison.
RESPONSES: Thanks for your reminder. As mentioned earlier, we have made a comparison before selecting the learning model. In this section, we only made optimization adjustments from the embedding dimension and time window two key parameters when setting hyperparameters. Based on the error analysis results, we finally comprehensively determined the key parameters in the AGCRN model of the model.
5. This application is a pure data-driven model; however, the authors did not give too much introduction to data mining with the data. The authors should add an introduction to the data interpolation and feature engineering.
RESPONSES: Thanks for your reminder. To ensure the data quality, it was necessary to preprocess the raw data of the the support working resistance. Outliers occurred during circumstances in which there were short or long intervals of data collection. In this study, the outliers of the support working resistance with a duration of less than 20 minutes or longer than 3 hours were eliminated. The missing values of the support working resistance are generally due to data loss during transmission or sensor errors. For the missing values, the Lagrange interpolation method was applied to fill the gaps. And the introduction to preprocess the raw data of the the support working resistance in line 200-206.
6. How do you ensure that you have ‘good’ data in your approach? Or more precisely: How do you ensure that no overfitting occurs? The results on both the training and validation dataset and learning curves are missing, which will help us evaluate if the data-driven model is overfitting. Please add in the revision.
RESPONSES: Thanks for your reminder. Your suggestion is valid. The on-site testing of the support working resistance data of the fully mechanized mining face is real-time collection, and it is mutually verified and analyzed with the theoretical analysis results of mine pressure and rock control. When predicting the support working resistance data, we not only focus on the results of the model training curve, but also incorporate the theoretical analysis results in the evaluation and analysis to avoid overfitting of the data.
7. The model evaluation is not clearly introduced. To prevent overfitting, a K-fold cross-validation technique should be adopted to help evaluate the model to avoid a high variance and bias in the results generated. Besides, one sampling comparison is not convincing enough, you need to show the average results of RMSE for a repeated sub-sampling process, that’s why repeated cross-validation must be added here. Please add in the revision.
RESPONSES: Thank you, your suggestion is valid. It can solve the problems on adaptability of prediction model more effectively. We have carefully evaluated the funding and resources required to complete these additional studies and found that such an expanded study is not currently affordable. Meanwhile, we feel that the scope of work of the present paper can support its conclusions. Therefore, we suggest that the additional experiments be included in a follow-up paper.
Reviewer 3 Report
Comments and Suggestions for Authors
Dear Editor Journal of applied science
The article entitled “An AGCRN Algorithm for Pressure Prediction in an Ultra-Long 2 Mining Face in a Medium-Thick Coal Seam in the Northern 3 Shaanxi Area, China”, have useful scientific information. But, paper in present format require to minor revisions. Specific remarks are as follow:
1- Important results should be expressed quantitatively in abstract.
2- In terms of grammar, paper should be re written again.
3-Quality of figures should be better in text.
4- Dependent and independent variables should be clearly shown in the text.
5- How much are the results obtained with the presented method different from analytical-numerical methods? Mentioned in the text of the article
Best regards
Comments on the Quality of English LanguageIn terms of grammar, paper should be re written again.
Author Response
1. Important results should be expressed quantitatively in abstract.
RESPONSES: Thanks for your advice. Important results were expressed in abstract part.
2.In terms of grammar, paper should be re written again.
RESPONSES: Thank you very much for finding this error. We are sorry for this grammar problem and have re written it according to your suggestion. In addition, we have asked a native English editor to revise the manuscript.
3. Quality of figures should be better in text.
RESPONSES: Thanks for your reminder. We have modified the relevant charts in the paper and improved their quality.
4. Dependent and independent variables should be clearly shown in the text.
RESPONSES: Thanks for your reminder. We apologize for not expressing ourselves clearly. We have revised the corresponding part of the manuscript and made to make the expression clearer and more accurate.
5.How much are the results obtained with the presented method different from analytical-numerical methods? Mentioned in the text of the article.
RESPONSES: Thanks for your reminder. We compared the predicted results with the field measured data and related literature, and found that the predicted model has good accuracy. In the next step, we will combine the numerical analysis results and continuously optimize the model parameters to expand the application range and effect of the model. And the plan and explanation were added in the text.